# Ultrafast transition from coherent to incoherent polariton nonlinearities in a hybrid 1L-WS$_2$/plasmon structure

Daniel Timmer [1,11], Moritz Gittinger [1,11], Thomas Quenzel[1], Alisson R. Cadore[2,7], Barbara L. T. Rosa[2], Wenshan Li[2], Giancarlo Soavi [2,8,9], Daniel C. Lünemann[1], Sven Stephan[1,10], Lara Greten[3], Marten Richter [3], Andreas Knorr [3], Antonietta De Sio [1,4], Martin Silies[1,10], Giulio Cerullo [5,6], Andrea C. Ferrari [2] & Christoph Lienau [1,4] ✉

Exciton polaritons based on atomically thin semiconductors are essential building blocks of quantum optoelectronic devices. Their properties are governed by an ultrafast and oscillatory energy transfer between their excitonic and photonic constituents, resulting in the formation of polaritonic quasiparticles with pronounced nonlinearities induced by the excitonic component. In metallic, plasmonic nanoresonators, dissipation phenomena limit the polariton lifetime to a few tens of femtoseconds, so short that the role of these polaritons for the nonlinear response of such hybrids is yet unexplored. Here we use ultrafast two-dimensional electronic spectroscopy (2DES) to uncover coherent polariton dynamics in a hybrid monolayer (1L) WS$_2$/plasmonic nanostructure. With respect to an uncoupled WS$_2$ flake, we observe an over 20-fold, polarization-dependent enhancement of the optical nonlinearity and a rapid evolution of the 2DES spectra within ~70 fs. We relate these dynamics to a transition from coherent polaritons to incoherent excitations, unravel the microscopic origin of the optical nonlinearities and show the potential of coherent polaritons for ultrafast all-optical switching.

Hybridization of electronic excitations with confined light modes is a powerful approach for manipulating optical and electronic properties in quantum materials[1–5]. Especially, plasmonic nanostructures allow for tailoring the coupling between excitonic emitters and localized optical near fields[1,6–8], enabling long-range[9,10] and plasmon-mediated energy transfer[11]. In such hybrids, subwavelength localization of the plasmonic field[1,6] and the resulting spread in momentum space enables coupling to momentum-dark excitons[12,13], which—owing to momentum mismatch—cannot be excited by far-field light. While the effects of such couplings on the linear optical spectra of plasmonic hybrids are broadly understood[12], their role for their optical nonlinearities has not been explained yet. This holds in particular for structures based on monolayer (1L) transition metal dichalcogenides (1L-TMDs)[14], semiconducting materials for which many-body interactions (MBIs) such as excitation-induced dephasing (EID) and resonance energy shifts (EIS) play a fundamental role for their optical nonlinearities[15–17]. In such

[1]Institut für Physik, Carl von Ossietzky Universität Oldenburg, Oldenburg, Germany. [2]Cambridge Graphene Centre, University of Cambridge, Cambridge, UK. [3]Institut für Physik und Astronomie, Nichtlineare Optik und Quantenelektronik, Technische Universität Berlin, Berlin, Germany. [4]Center for Nanoscale Dynamics (CENAD), Carl von Ossietzky Universität Oldenburg, Oldenburg, Germany. [5]Dipartimento di Fisica, Politecnico di Milano, Milan, Italy. [6]Istituto di Fotonica e Nanotecnologie, CNR, Milan, Italy. [7]Present address: Brazilian Nanotechnology National Laboratory, Brazilian Center for Research in Energy and Materials, São Paulo, Brazil. [8]Present address: Institute of Solid State Physics, Friedrich Schiller University Jena, Jena, Germany. [9]Present address: Abbe Center of Photonics, Friedrich Schiller University Jena, Jena, Germany. [10]Present address: Institute for Lasers and Optics, University of Applied Sciences, Emden, Germany. [11]These authors contributed equally: Daniel Timmer, Moritz Gittinger. ✉e-mail: christoph.lienau@uni-oldenburg.de

systems, strong coupling to plasmons may result in a hybridization of bright and momentum-dark excitons. It is not obvious, however, how this coupling competes with MBIs or how it affects the polariton dynamics. For spin bright and dark excitons in 1L-WSe₂, such a hybridization can be induced by external magnetic fields[18].

Previous pump–probe studies on 1L-WS₂ on plasmonic Ag nanostructures[19–22] reported enhanced optical nonlinearities[19] and ultrafast modulation of the coupling strength[20]. However, the time resolution was insufficient to explore their coherent dynamics[11,23,24]. Thus, it is crucial to experimentally explore coherent energy transfer processes, specifically Rabi oscillations[11,24,25] that are direct markers for exciton–plasmon hybridization in 1L-TMD/metal structures. In particular, studies probing coherent polariton excitations and their decay into incoherent excitations are needed[26–28].

Ultrafast two-dimensional electronic spectroscopy (2DES)[29,30], with a time resolution shorter than the polariton dephasing time, is a powerful method to follow coherent and incoherent flow of energy in hybrid quantum systems[11,24,26,31–36]. 2DES replaces the impulsive excitation in pump–probe spectroscopy with two phase-locked ultrashort excitation pulses[29]. This creates 2DES maps that correlate excitation and detection energies of the system as a function of the waiting time $T$ between the second pump and the time-delayed probe. Oscillations of the cross and diagonal peaks in these maps provide distinct signatures of Rabi oscillations and, thus, coherent energy transfer dynamics[11,24,27]. 2DES studies of such coherent dynamics have mainly been performed for J-aggregated molecules coupled to plasmonic resonators[11,24]. A 2DES study of 1L-WSe₂ embedded in a microcavity revealed multiple polariton branches arising from exciton–photon–phonon hybridization[35]. The signatures of coherent polariton excitations in 2DES spectra have been the focus of a series of recent theoretical works[37–42].

Here we use 2DES with a 10 fs time resolution to explore the optical nonlinearities of a hybrid nanostructure comprising 1L-WS₂ on a silver nanoslit array at room temperature. We observe that exciton–plasmon hybridization results in a 20-fold increase in optical nonlinearity compared with the uncoupled 1L-WS₂ and record distinct changes in the 2DES lineshape within the first ~70 fs, marking the transition from coherent to incoherent polaritons and long-lived dark states. We rationalize these observations by considering the hybridization of plasmons with bright and dark excitons and the effects of EID and Pauli blocking on the polariton resonances. Supported by theoretical modelling, our results distinguish between coherent and incoherent polaritons in 2DES, shedding light on their nonlinear response.

## Hybrid 1L-TMD/plasmonic nanostructures

We fabricate a periodic nanoslit array in a polycrystalline silver film (Fig. 1a). The slits act as periodic perturbations of the metal surface, allowing for far-field excitation of surface plasmon polaritons (SPPs)[6,11,25]. A single layer of WS₂ is dry-transferred on top of the 20 × 50 µm² nanoslit array (Supplementary Fig. 1). The array is designed to tune the SPP in resonance with the A exciton of 1L-WS₂ ($E_X \approx 2$ eV)[23] at an angle of incidence close to $\theta = 6°$ (Fig. 1b). The array is characterized by a negative SPP dispersion[6,11] and a vanishing optical nonlinearity (Supplementary Fig. 18). Figure 1c compares the photoluminescence (PL) spectrum of 1L-WS₂ (red) and linear reflectivity of the hybrid array at $\theta = 3°$ (blue). The dipolar coupling between excitons and SPPs leads to the formation of new upper (UP) and lower (LP) polariton resonances. Angle-dependent reflectivity spectra (Fig. 1b) show an avoided crossing between the UP and LP branches, well reproduced by finite difference time domain (FDTD) simulations (Supplementary Section 3). We deduce an exciton–SPP coupling strength $V_{XP} \approx 24$ meV, while the dephasing rates of excitons and SPP are $\gamma_X = 19$ meV and $\gamma_{SPP} = 34$ meV, respectively. This places the sample at the border between the intermediate and strong coupling regime, reached at $V_{XP} > \sqrt{(\gamma_X^2 + \gamma_{SPP}^2)/2}$ (refs. 1,6) (Supplementary Section 4).

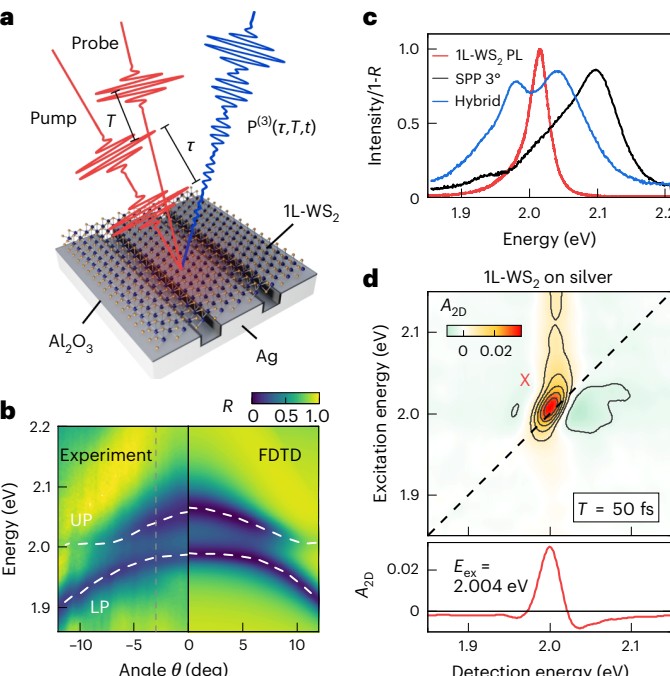

**Fig. 1 | Hybrid 1L-TMD/plasmonic nanostructure. a**, Sample geometry (not to scale) and experimental pulse sequence. The plasmonic Ag nanoslit array (slits with 495 nm period, 45 nm width and 45 nm depth) is coated with 5 nm Al₂O₃ and covered with 1L-WS₂. Ultrafast pump–probe and 2DES are recorded in reflection at an incidence angle $\theta$ using an excitation pulse pair with delay $\tau$, delayed with respect to a third probe pulse by $T$. The nonlinear signal (blue) is emitted into the reflected probe direction (not shown) and recorded by a spectrometer. **b**, Linear $\theta$-dependent reflectivity, showing hybridization between the 1L-WS₂ A exciton (~2 eV) and the angle-dependent SPP, forming upper (UP) and lower (LP) polariton resonances with a crossing angle at $\theta_C \approx 6°$. **c**, PL of 1L-WS₂ A exciton close to 2 eV (red) and SPP absorption (1-$R$) at $\theta = 3°$ (black). In the presence of the 1L-TMD, the SPP red-shifts ~90 meV owing to the change of dielectric function at the interface. The hybrid nanostructure (blue) shows UP and LP peaks. **d**, 2DES map for 1L-WS₂ A excitons on a bare Al₂O₃-coated Ag film. The weak nonlinear signal is dominated by MBIs, in particular EID.

To investigate the ultrafast polariton dynamics, we perform pump–probe and 2DES experiments with a 10 fs time resolution[11,23,43] using a home-built non-collinear optical parametric amplifier[23], operating at 175 kHz. Pump and probe pulses with spectra covering 520–700 nm (Supplementary Fig. 10) are focused to spot sizes <20 µm under the same angle $\theta$. The reflected probe spectrum is recorded as a function of detection energy $E_{det}$, yielding differential reflectivity $\Delta R/R$ spectra. 2DES measurements are performed in a partially collinear geometry, using a phase-stable excitation pulse pair (Fig. 1a). 2DES maps are obtained by scanning $\tau$ and subsequent Fourier transform[29] (Methods).

A 2DES map recorded at $T = 50$ fs for 1L-WS₂ is shown in Fig. 1d. Its main feature is the positive A-exciton diagonal peak close to 2 eV. Along $E_{det}$, the spectrum shows negative side lobes, asymmetrically centred around the exciton. This lineshape can be explained using the current understanding of the exciton nonlinearity of 1L-TMDs[16,17,23]. The 2DES lineshape[23,44] results from the overlap of positive ground state bleaching (GSB) and stimulated emission (SE) signals, probing the transition from the ground state to the exciton $|X⟩$, and negative excited state absorption (ESA) from $|X⟩$ to unbound two-exciton states $|XX⟩$ (refs. 23,30,45). In 1L-TMDs, MBIs[16,17,23,30,46] result in a spectral shift (EIS) and an increase in damping[15] (EID) of the two-exciton relative to the exciton transition. For EID, excitonic linewidth broadening introduces characteristic negative side lobes in $\Delta R/R$ spectra (Fig. 1d and Supplementary Fig. 11). Lineshape asymmetries arise from finite EIS (Supplementary Fig. 11). For 1L-WS₂, weak effects of Pauli blocking,

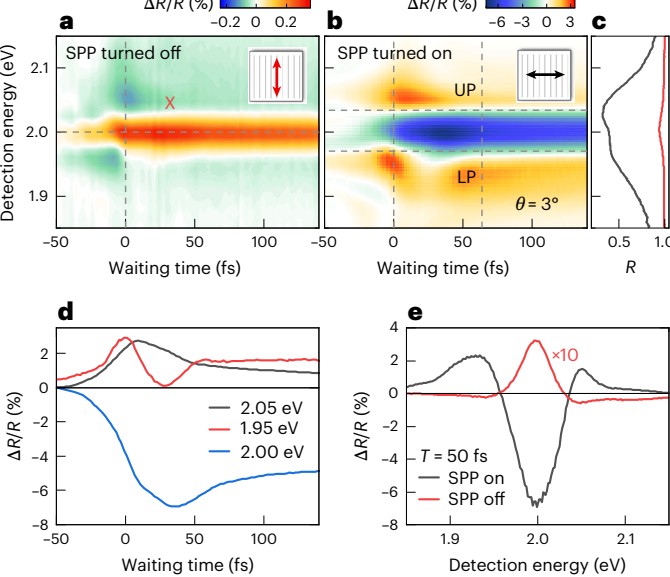

**Fig. 2 | Anisotropic plasmon hybridization and optical nonlinearity in 1L-WS₂ nanoslit arrays. a**, Pump–probe map recorded with polarizations parallel to the nanoslits with a pump fluence of 10 μJ cm⁻². Since SPP excitation is effectively turned off, the weak nonlinear signal mainly probes the 1L-WS₂ response. The positive exciton resonance around 2 eV is dominated by EID and similar to the reference measurement on a bare substrate. **b**, Pump–probe map for $\theta = 3°$, slightly above the crossing angle (6°), for polarizations perpendicular to the nanoslits with the same pump fluence. The $\Delta R/R$ spectra exhibit a strong polariton nonlinearity with a negative ESA signal close to 2 eV and positive UP and LP bands. Oscillatory dynamics appear with a period comparable to the Rabi period of $T_R \approx 64$ fs (vertical lines) deduced from linear spectroscopy. **c**, Linear reflectivity spectra for polarization parallel (red) and perpendicular (black) to the nanoslits. **d**, Crosscuts taken at the UP (black), LP (red) and ESA (blue) band in **b**, emphasizing the transient, oscillatory modulation. **e**, Spectral crosscuts at $T = 50$ fs from **a** (red) and **b** (black), highlighting the 20-fold enhancement of the nonlinearity induced by X–SPP coupling.

resulting in a decrease of the exciton oscillator strength upon excitation, are less pronounced than EID and EIS[16,17,23,46]. A slight tilting of the A exciton peak indicates a small inhomogeneous broadening owing to local strain and disorder of the monolayer[15,47].

## Enhancement of optical nonlinearity

Since SPPs of the nanoslit array can only be excited with *p*-polarized light, perpendicular to the grating lines, selective probing of the X–SPP coupling is achieved by changing the laser polarization.

For *s*-polarization, $\Delta R/R$ maps at $\theta = 3°$ (Fig. 2a) are similar to those recorded for the bare 1L-WS₂, in the absence of the grating[23]. Here effects of the X–SPP coupling are effectively turned off. EID dominates the nonlinearity[17,23], and the spectra do not show substantial temporal evolution. By contrast, for *p*-polarized excitation and probing, X–SPP coupling profoundly affects the pump–probe spectra (Fig. 2b). We observe a >20-fold enhancement of $\Delta R/R$ (Fig. 2e). Now, $\Delta R/R$ spectra appear with a negative sign (ESA) around the exciton resonance and zero crossings at the energies of the UP and LP resonances in the linear spectra (Fig. 2c). Positive bands are seen above and below the UP and LP energies, respectively. As discussed below, these changes in lineshape can be understood by a transfer of MBIs from the TMD excitons to the hybrid system. Oscillatory modulations of these nonlinearities with ~60 fs period are seen within the first 100 fs after excitation. This is emphasized in Fig. 2b by marking the Rabi period $T_R = 2\pi\hbar/(E_{UP} - E_{LP}) \approx 64$ fs, deduced from the linear spectra, by vertical dashed lines. Cross-sections of the pump–probe map in Fig. 2d show the transient dynamics at different detection energies.

Such an enhancement of the optical nonlinearity is particularly striking when considering the lack of nonlinearity of the bare nanoslit array in the absence of the excitonic material (Supplementary Fig. 18). By merely adding 1L-TMD, the amplitude in $\Delta R/R$ increases up to a maximum of 10%, a nonlinearity that exceeds that of the excitonic constituent by far. Large optical nonlinearities have also been observed in the transient spectra for 1L-WS₂ covered with Ag nano-disk arrays[19], yet without reaching the regime of coherent polariton dynamics. In contrast to 1L-TMD-based systems, hybrids using molecular J-aggregates with large transition dipole moments[6,43] so far showed much reduced enhancements, with optical nonlinearities comparable to those of the uncoupled excitonic system[11].

## Ultrafast 2DES

To gain more insight into the microscopic origin of the enhanced nonlinearity and rapid spectral evolution, we perform 2DES to correlate excitation and detection pathways of the hybrid system. 2DES maps for selected waiting times are presented in Fig. 3a–d. For $T = 0$ fs, the map shows four distinct positive peaks in the energy region of the polariton resonances (peaks 1–4; Fig. 3a). These are surrounding a central negative ESA feature (peak 5) with a lineshape slightly tilted along the diagonal. Importantly, the off-diagonal peaks 1 and 4 appear as positive peaks, centred close to the energies of the ($E_{ex} = $ UP, $E_{det} = $ LP) and (LP, UP) cross peaks (dashed lines, Fig. 3a). Peaks 2 and 5 are centred around the (UP, UP) diagonal, while 3 and 5 around (LP, LP). We interpret each of these pairs as a diagonal peak with dispersive lineshape. Such a combination of dispersive diagonal and absorptive cross peaks is detected during the first 30 fs (Supplementary Fig. 17), before the 2DES lineshape undergoes a rapid evolution. Within 70 fs, it changes into a stripe-like pattern with two vertically oriented positive stripes (Fig. 3d), centred around a negative ESA stretched along the excitation axis. The rapid and partly oscillatory evolution of the 2DES spectra is shown in Fig. 3e for peaks 1–5. This evolution marks the transition from impulsively excited, short-lived coherent polaritons to incoherent polariton excitations and long-lived (>20 ps; Supplementary Fig. 19) dark states, demonstrated here for a 1L-TMD/metal hybrid.

## Interpretation of 2DES maps

We base our interpretation on a minimal three-coupled oscillator model (3-COM)[12,48], recently introduced to describe linear optical properties of 1L-TMD/plasmon hybrids. A key ingredient is the coupling of plasmonic modes to both momentum-bright and momentum-dark excitons that lie outside of the light cone[12,48]. In the case of 2D nano-disk arrays, strong spatial near-field localization results in strong coupling to dark 1L-TMD excitons, while far-field coupling to bright excitons is much reduced[12]. For our nanoslit array, we estimate a coupling of plasmonic modes to bright and dark excitons of similar magnitude (Supplementary Section 8). We therefore analyse a 3-COM[12,48] that includes 3 harmonic oscillators to account for a bright plasmon ($|P\rangle$) coupled to the bright ($|X_B\rangle$) and dark ($|X_D\rangle$) excitons, with coupling strengths of 19 meV each. Their energies are set to $E_{X_D} = E_{X_B} = 2$ eV, and the SPP is slightly blue-shifted to $E_P = 2.015$ eV, placing us in the regime of small detuning where $|X\rangle$ and $|P\rangle$ are almost in resonance. Importantly, optical absorption of the SPP resonance $|P\rangle$ is much stronger than that of $|X_B\rangle$. This is introduced in 3-COM by choosing an SPP transition dipole moment 10 times larger than that of $|X_B\rangle$, while $|X_D\rangle$ is assumed to be dark.

We now map this 3-COM onto an effective Hamiltonian and analyse the time evolution of the density matrix of the coupled system on the basis of the Lindblad master equation[11,23] (Methods and Supplementary Section 7).

As a result of X–SPP coupling, we obtain bright $|UP\rangle$ and $|LP\rangle$ polaritons with 60 meV normal mode splitting. In addition, a third, almost dark state $|D\rangle$ appears at $E_X$ (Fig. 4a). This dark state is taken as a simplistic model for the entire reservoir of dark states in the hybrid system. Since $|D\rangle$ is a superposition of bright and dark excitonic states

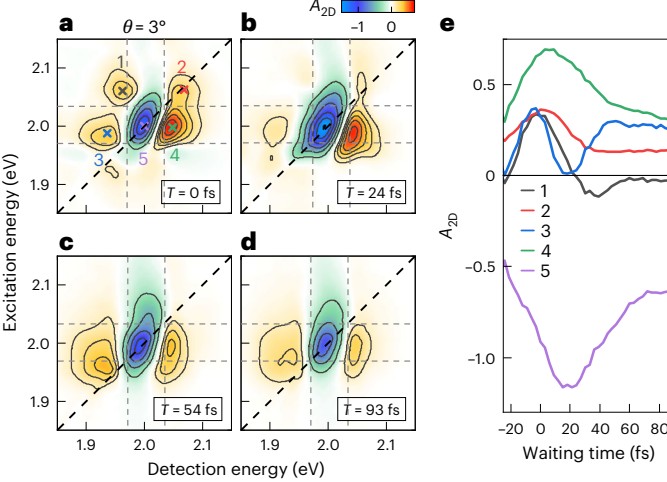

**Fig. 3 | Ultrafast 2DES of hybrid 1L-WS$_2$ nanoslit array for $\theta$ = 3°. a–d**, Maps for selected waiting times of 0 fs (**a**), 24 fs (**b**), 54 fs (**c**) and 93 fs (**d**), revealing a rapid transformation of the 2DES spectra during polariton dephasing. At $T$ = 0 fs (**a**), 4 dominant positive peaks (1–4) appear in the vicinity of a strong ESA band close to 2 eV (peak 5, tilted along the diagonal). Peaks 1 and 4 are polariton cross peaks, while 2 and 3 are parts of UP and LP diagonal peaks with dispersive lineshape. UP and LP energies are marked by dashed lines. Within the first 70 fs, the spectra undergo rapid evolution until only a featureless vertically oriented stripe structure remains (**c,d**). This change in 2DES lineshape marks the transition from coherent to incoherent polaritons. **e**, Waiting time dynamics of the 2DES peaks 1–5 taken at the positions marked in **a** within a ±5 meV window. Pronounced temporal modulations appear during the first 70 fs.

with zero plasmon admixture, its oscillator strength is much smaller than that of |UP⟩ and |LP⟩ (Supplementary Section 4). The linear optical spectrum is therefore dominated by the two polariton resonances with almost equal oscillator strength (Fig. 4b, inset), consistent with the linear spectra in Fig. 1c.

In this case, impulsive optical excitation mainly couples to the plasmon resonance |P⟩ and induces coherent energy transfer between |P⟩ and both |X$_B$⟩ and |X$_D$⟩ (Fig. 4b), giving rise to strongly damped Rabi oscillations with an ~65 fs period. In the polariton basis (Fig. 4c), this corresponds to similar initial populations of |UP⟩ and |LP⟩. Both populations decay rapidly, within <50 fs, owing to the radiative damping of the plasmonic mode, included in the Lindblad equation[49]. In addition, exciton dephasing contributes to polariton decoherence and induces polariton population transfer into the dark |D⟩ state within <50 fs (refs. 39,50). In our model, the |D⟩ population decays mainly owing to dephasing-induced back transfer of population into the radiatively damped polaritons[39,50]. A recent theoretical study of molecular cavity polaritons also suggests that polariton decoherence arises from population transfer into collective dark states, mediated by molecular vibrations[51].

While linear optical experiments solely probe transitions from the ground state to the singly excited or one-quantum (1Q) polariton states, transitions between 1Q and the manifold of two-quantum states (2Q) contribute to the nonlinear experiments. In third-order perturbation theory, these give access to the second rank of the many-body Hamiltonian, while higher ranks cannot be reached[11,39,52–54]. An intuitive and powerful approach for understanding nonlinear spectra therefore considers the interplay between GSB/SE transitions and ESA pathways, while phenomenologically introducing MBIs by modifying the properties of the 2Q states[23,30,44,52,53] (Supplementary Section 7). We follow this approach and include doubly excited plasmon |2P⟩, bright and dark two-exciton states |XX$_B$⟩ and |XX$_D$⟩ and three mixed 2Q states |P, X$_D$⟩, |P, X$_B$⟩ and |X$_B$, X$_D$⟩ in the 3-COM. In the polariton basis (Fig. 4a), this gives rise to doubly excited |2UP⟩ and |2LP⟩ and mixed |UPLP⟩

polaritons, a doubly excited dark state |2D⟩ and mixed dark-state polaritons |DUP⟩ and |DLP⟩. In the absence of MBIs, this extended 3-COM produces a Hamiltonian with vanishing third-order nonlinearity. For our experimental conditions, the SPP nonlinearity is negligible (Supplementary Fig. 18); hence, the plasmonic mode remains a linear oscillator. Thus, the nonlinearity of the hybrid system solely arises from its excitonic constituents. In our model, EID is introduced via a slight increase in the pure dephasing rate of the |X⟩ → |XX⟩ ESA. EIS, by contrast, results from slight energetic shifts of the two-exciton states. Pauli blocking manifests itself in a reduction of the transition dipole moment of the |X⟩ → |XX⟩ transition compared with |0⟩ → |X⟩ (refs. 16,26,46).

Since EID has a dominant role for the excitonic nonlinearity for 1L-WS$_2$ (refs. 15–17,23), we discuss its effects on the hybrid system. At early waiting times, before onset of incoherent relaxation, only |LP⟩ and |UP⟩ are optically excited, while the |D⟩ state remains essentially unpopulated. Owing to EID, the linewidth of the optically allowed transitions between |UP⟩ and |LP⟩ and the 2Q states (level scheme in Fig. 4a) is slightly larger than that of the GSB/SE band. In the strong coupling regime, this results in 2DES maps showing four polariton diagonal and cross peaks with characteristic EID lineshape, that is, one positive central peak with two symmetric, negative sidelobes along $E_{det}$. In our regime, near the onset of strong coupling, the negative ESA contributions of these peaks overlap spectrally, resulting in a vertically oriented stripe-like pattern (Fig. 4e). With increasing waiting time, the upper and lower polaritons are rapidly damped, partly irreversibly by their radiative decay via the plasmonic antenna, partly reversibly by an exciton dephasing-induced transfer to the dark states (Fig. 4b). This relaxation into dark states opens new ESA pathways from |D⟩ to the mixed |DUP⟩ and |DLP⟩ states. Since EID also increases the dephasing rate of these new ESA transitions over those of the 1Q transitions, this again results in vertically striped lineshapes of the 2DES maps, very similar to those at early waiting times (Fig. 4g). The amplitude of the map, however, decreases with the 1Q-state lifetime. While the shape of the simulated 2DES map agrees with that seen experimentally at later waiting times (Fig. 3c,d), it fails to reproduce the 2DES maps at early waiting times (Fig. 3a). Even though EID may contribute to the polariton nonlinearity, other physical mechanisms appear more dominant during the polariton coherence time.

The effects of EIS on 2DES of strongly coupled polaritons have recently been investigated for J-aggregate/metal hybrids, where blue-shifts of the two-exciton states dominate the nonlinear response[11]. In that case, EIS gives rise to distinct dispersive lineshapes of all polariton diagonal and cross peaks, qualitatively different from those seen in Fig. 3a. Our data therefore indicate that EIS plays a minor role for the nonlinearity of the TMD/plasmon hybrid.

We now turn to the effects of Pauli blocking[16] on the 2DES maps. Intuitively, the creation of excitons in the 1L-TMD by optical pumping may be expected to reduce the probability for coherent energy transfer between plasmons and excitons, by enhancing SE from excitons, while reducing their absorption. Pauli blocking is therefore expected to reduce the polariton normal mode splitting. Such 'contraction' of the Rabi energy[52] has been observed experimentally for strong pumping of J-aggregates[55] and 1L-TMDs[19] coupled to plasmonic nanostructures. Rabi contractions have been used to explain molecular polaritons[39,52,56] and to model power-dependent spectra of TMD-based polaritons[5,57,58].

In our 3-COM, the Pauli blocking reduction of the |X⟩ → |XX⟩ transition dipole moment introduces energy shifts of the 2Q states in the polariton basis. Specifically, it reduces the energy splitting between |2UP⟩ and |2LP⟩, as well as between |DUP⟩ and |DLP⟩ (Fig. 4a). The energies of the mixed |UPLP⟩ and |2D⟩ states, however, remain unchanged. This has important consequences for the lineshape of the 2DES maps during the coherence time. For the diagonal polariton peaks, these lineshapes are defined by the spectral properties of the |GS⟩ → |UP(LP)⟩ and |UP(LP)⟩ → |2UP(2LP)⟩ transitions. The energy shifts of the 2Q states therefore result in dispersive lineshapes of the diagonal peaks along

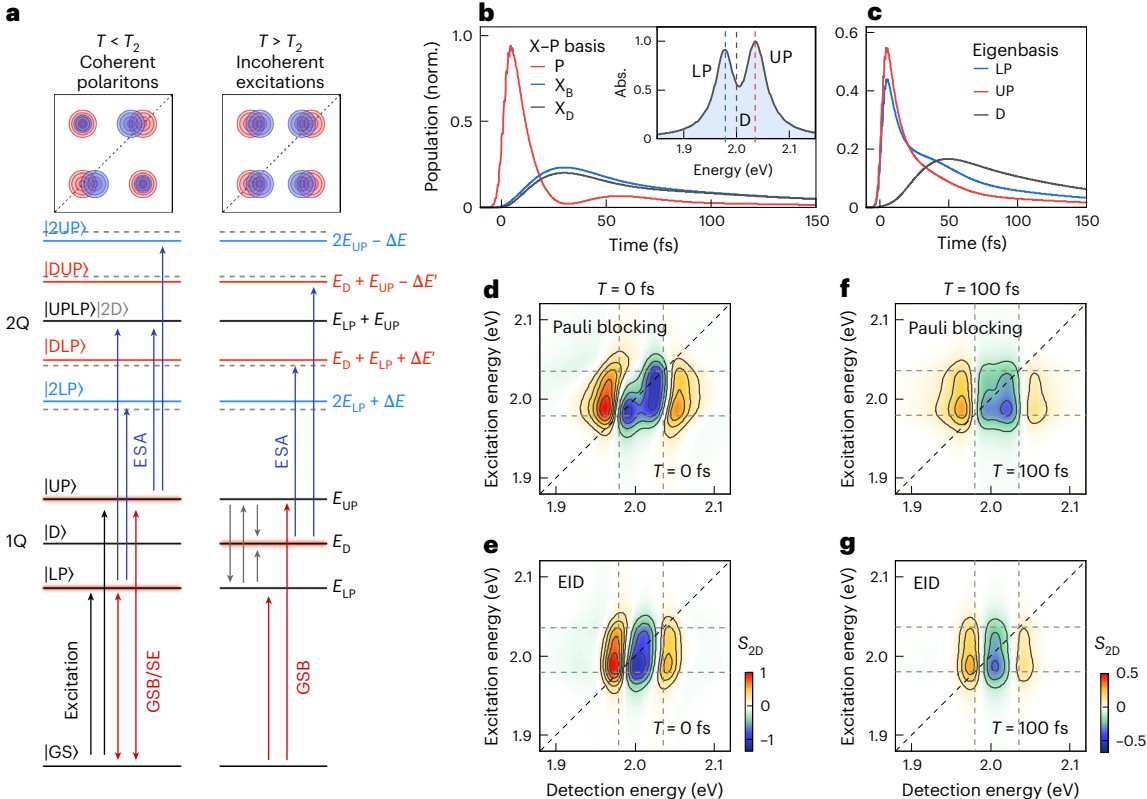

**Fig. 4 | From coherent to incoherent polariton nonlinearities in 1L-WS$_2$/plasmon hybrids. a**, Eigenstate structure of the 3-COM Hamiltonian. Bright ($|UP\rangle$, $|LP\rangle$) and dark ($|D\rangle$) polaritons result from the coupling of dark and bright excitons to the SPP resonance. Optical excitation mainly drives UP and LP resonances (left). The probe induces (positive) GSB/SE signals (red arrows) in 2DES (red peaks in the top left scheme). Transitions to the 2Q manifold (blue arrows) lead to negative ESA in 2DES (blue peaks). Finite Pauli blocking reduces the normal mode splitting in the 2Q manifold (Rabi contraction). In 2DES, this results in dispersive diagonal peaks while the cross peaks, probing ESA to $|UPLP\rangle$, have absorptive lineshapes with positive amplitude. Relaxation into dark $|D\rangle$ states induces new ESA transitions and alters the lineshape of the 2DES maps,

giving rise to dispersive diagonal and cross peaks (top right scheme). **b,c**, Normalized (norm.) population dynamics in the uncoupled X–P (**b**) and polariton eigenbasis (**c**) and linear absorption (Abs.) spectrum (inset). $|X\rangle$ and $|P\rangle$ show faint Rabi oscillations. $|UP\rangle$ and $|LP\rangle$ populations (red and blue) decay quickly into longer-living dark states (black). **d,e**, Simulated 2DES maps for the regime of coherent polaritons at $T = 0$ fs for Pauli blocking nonlinearity (**d**) and EID (**e**). $|UP\rangle$ and $|LP\rangle$ energies are marked with dashed lines. Absorptive cross peaks as a signature of Rabi contractions appear in **d**. **f,g**, Simulated 2DES maps for the regime of incoherent excitations at $T = 100$ fs for Pauli blocking nonlinearity (**f**) and EID (**g**). In both cases, all 2DES peaks show a dispersive lineshape.

$E_{det}$. Since Pauli blocking downshifts $|2UP\rangle$ while upshifting $|2LP\rangle$, the signs of the two dispersive peaks are opposite (Fig. 4d). By contrast, for the cross peaks, ESA involves the transitions from $|UP(LP)\rangle$ to the spectrally unshifted mixed $|UPLP\rangle$ state. Thus, the cross peaks have an absorptive lineshape with a positive sign resulting from the reduced dipole moment of the ESA (Fig. 4d). Qualitatively, the shape of experimental 2DES maps in the coherent polariton regime (Fig. 3a) agrees well with that simulated for a Pauli blocking nonlinearity. Pauli blocking can also rationalize the rapid, <70 fs, evolution of the 2DES spectra. With increasing loss of polariton coherence, the role of dark states becomes more important, inducing ESA to the mixed $|DUP\rangle$ and $|DLP\rangle$ states. Since Pauli blocking introduces energetic shifts of these mixed states, all four polariton peaks appear with dispersive lineshapes, resulting in a vertically striped 2DES map (Fig. 4f), consistent with maps measured at later waiting times (Fig. 3c,d) and simulated with our EID model. The temporal evolution in the shape of the 2DES maps during the first 70 fs in Fig. 3 thus is the characteristic signature for the transition from a regime of coherent polaritons to one of incoherent polaritons and dark state excitations. Oscillations in the amplitude of the 2DES map during this period (Fig. 3e) reflect UP/LP quantum beats launched by the pump pulses and are the markers for a coherent flow of energy between plasmons and 1L-TMD excitons in the coupled 1L-TMD/metal hybrid. Beyond this initial coherent regime, the nonlinearities of the hybrid structure are induced by incoherent polaritons and dark states

and time-resolved spectra monitor the relaxation and decay of these incoherent excitations.

## Conclusion

Taken together, our 2DES study reveals a regime of coherent ultrafast polariton nonlinearities in 1L-TMD/metal hybrids. When covering an Ag nanoslit array with a 1L-TMD, transient changes in sample reflectivity by up to 10% are observed on a tens of fs timescale. Such large coherent polariton nonlinearities, likely to be increased further by optimizing the plasmonic nanoresonator, are of substantial interest for efficient, ultrafast switching of light by light on the nanoscale. Recently, femtosecond switching of TMD monolayers strongly coupled to photonic microcavities has been demonstrated with record-low fluences in the few-pJ regime[5]. In these experiments, an increase in switching efficiency has been reached by exploiting incoherent dark excitons to transiently reduce the polariton normal mode splitting. The use of incoherent excitons limits the achievable switching time to a few 100 fs. Our experiments now demonstrate how to increase the switching time by at least an order of magnitude by using fully coherent polariton nonlinearities. For exploiting this potential, the suppression of nonlinearities induced by incoherent dark polaritons is key. For this, strong coupling to TMD heterolayers may be of interest, since such heterolayers can provide ultrafast exciton relaxation channels into trap states[59] that spatially separate dark excitons from the region of strongly coupled excitons,

suppressing incoherent excitonic nonlinearities. When combined with suitable metallic or dielectric nanoresonators, this may offer a route towards active nanostructures and metasurfaces with ultrafast response times. Spatially resolved studies of the ultrafast dynamics of such switching devices are key to uncovering this potential.

## Online content

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

## Methods

### Sample preparation

Polycrystalline 200-nm-thick Ag films are deposited on a fused silica substrate using electron beam physical vapour deposition. Ga-based focused ion beam milling is used to fabricate plasmonic nanoslit arrays with a size of $20 \times 50\ \mu m^2$ (45 nm slit width and depth and 495 nm period). The film containing the slit arrays is coated with a 5-nm-thick layer of aluminium oxide grown at 150 °C. 1L-WS$_2$ flakes are prepared by micro-mechanical exfoliation on Nitto Denko tape[60] from bulk 2H-WS$_2$ (HQ Graphene source) onto polydimethylsiloxane stamp. Selected flakes are aligned and stamped onto the nanoslit arrays with an $x,y,z$ micro-manipulators at 60 °C (ref. 61). One array covered with an ~$50 \times 70\ \mu m^2$ flake is chosen for all experiments. Raman and PL spectra are recorded to confirm the number of layers[62] (see Supplementary Section 1 for more details).

### Ultrafast pump–probe and 2DES

Experiments are performed with a home-built set-up[43] using 9 fs pulses (520–700 nm) generated with a non-collinear optical parametric amplifier[23,63], pumped by a fibre amplifier system (Tangerine V2, Amplitude Systems) operating at 175 kHz. These spectra are used both as pump and probe in the 2DES set-up. A phase-stable pump pulse pair with delay $\tau$ (coherence time) is generated using an in-line birefringent interferometer (TWINS)[64]. A relative delay of the probe in respect to the second pump pulse (waiting time $T$) is set via a motorized retroreflector (M126.DG, Physik Instrumente). Pump and probe are focused onto the sample to <20 µm spot size using an off-axis parabolic mirror. The reflected probe beam is recollimated and sent to a grating spectrograph (Acton SP 2150i) with a fast and sensitive line camera (Aviiva EM4, e2v). Differential reflectivity spectra

$$\frac{\Delta R}{R}(\tau, T, E_{det}) = \frac{S_{on}(\tau, T, E_{det}) - S_{off}(E_{det})}{S_{off}(E_{det})} \tag{1}$$

are recorded as a function of detection energy $E_{det}$ for probe spectra with a blocked ($S_{off}$) and unblocked ($S_{on}$) pump. For this, an optical chopper is used with a custom 500 slot wheel. Pulses are chopped in pairs at 43.75 kHz, and the camera records spectra at 87.5 kHz. For 2DES, $\tau$ is scanned (−50 fs to 160 fs) and 2DES maps are obtained after Fourier transform of such a coherence time scan via[65]

$$A_{2D}(E_{ex}, T, E_{det}) = \Re\left(\int_{-\infty}^{\infty} \Theta(\tau)\, \frac{\Delta R}{R}(\tau, T, E_{det})\, e^{iE_{ex}\tau/\hbar} d\tau\right) \tag{2}$$

where $\Theta(\tau)$ denotes the Heaviside step function and $\hbar$ is Planck's reduced constant. This yields absorptive 2DES maps as a function of $E_{det}$, $E_{ex}$ and $T$. Pump–probe data are recorded setting $\tau = 0$ (see Supplementary Section 6 for more details).

### Density matrix simulations

Time-dependent simulations of the density matrix $\hat{\rho}(t)$ and 2DES signals[11,23] are performed based on a non-perturbative approach that uses the Lindblad master equation[66,67], accounting for all field interactions while also including system-bath interactions via Lindblad operators $\hat{L}_k$. The master equation in Lindblad form[66,67]

$$\dot{\rho} = -\frac{i}{\hbar}[\hat{H}, \hat{\rho}] + \frac{1}{2}\sum_k (2\hat{L}_k \hat{\rho} \hat{L}_k^\dagger - \hat{L}_k^\dagger \hat{L}_k \hat{\rho} - \hat{\rho} \hat{L}_k^\dagger \hat{L}_k) \tag{3}$$

is solved numerically while accounting for all 3 laser electric fields using Gaussian-shaped laser pulses with a full-width at half maximum of the intensity profile of 5 fs, centred at 2.15 eV. Here $\hat{H} = \hat{H}_S + \hat{H}_I$ contains the system Hamiltonian $\hat{H}_S$ and light–matter interaction Hamiltonian $\hat{H}_I = -\hat{\mu}E(t)$, where $E(t)$ is the total laser electric field and $\hat{\mu}$ is the transition dipole moment operator. To simulate the experiments, the optical response is obtained by calculating the expectation value of the transition dipole moment operator along the detection time $t$ for fixed $\tau$ and $T$. Using a 4-step phase-cycling scheme, the nonlinear third-order polarization is then isolated from the total signal and 2DES maps are obtained by varying $\tau$ and $T$ accordingly and performing a Fourier transform along $t$ and $\tau$.

Exciton plasmon coupling based on the 3-COM is implemented using the plasmon P and two kinds of exciton X = {X$_D$, X$_B$}. The system Hamiltonian reads

$$\hat{H}_S = E_P \hat{b}_P^\dagger \hat{b}_P + \sum_X E_X \hat{b}_X^\dagger \hat{b}_X + V_{XP}\left(\hat{b}_X^\dagger \hat{b}_P + \hat{b}_P^\dagger \hat{b}_X\right) \tag{4}$$

using exciton creation and annihilation operators $\hat{b}_X^\dagger$ and $\hat{b}_X$, respectively, with A exciton energy $E_X = 2$ eV for both the momentum bright and dark excitons X$_B$ and X$_D$. Here we assume for simplicity that only dark states with small momenta close to the bottom of the dispersion relation are contributing since we are in the case where X$_B$ is almost in resonance with P. For the plasmon, we use the operators $\hat{b}_P^\dagger$ and $\hat{b}_P$ and plasmon energy $E_P = 2.015$ eV, which is estimated from the angle-dependent reflectivity of the hybrid structure. Since for $\theta = 3°$ we are slightly above the crossing angle ($E_P > E_X$), the plasmon is slightly blue-shifted relative to the excitons. X–SPP coupling is accounted for by the light–matter coupling in the rotating wave approximation[11] with coupling strengths $V_{XP}$. In addition to the ground state, both 1Q and 2Q states are considered in the simulation. The 1Q states are $|P\rangle$, $|X_B\rangle$ and $|X_D\rangle$. The 2Q states are doubly excited plasmon $|2P\rangle$, two-exciton states $|XX_B\rangle$ and $|XX_D\rangle$, and the mixed states $|P, X_D\rangle$, $|P, X_B\rangle$ and $|X_B, X_D\rangle$. Plasmon relaxation and exciton dephasing are accounted for via respective Lindblad operators $\hat{L}_k$ (ref. 11). Since this 3-COM implementation results in a linear Hamiltonian and does not produce a nonlinear response, we introduce a modification to account for effects owing to EID or Pauli blocking. For EID, the exciton dephasing for the transition from the $|X\rangle$ to $|XX\rangle$ state is increased by 10% in the Lindblad formalism. For Pauli blocking, the transition dipole moment for the $|X\rangle$ to $|XX\rangle$ transition is reduced by 1%. As a consequence, the 2Q coupling elements associated with the $|XX\rangle$ states are also reduced by this amount. Very similar 2DES maps as obtained from the 3-COM were reported for a molecular Tavis–Cummings model[39], considering the collective coupling of $N$ identical emitters to a single cavity mode (see Supplementary Information for a comparison) and for a plasmonic Jaynes–Cummings model[42]. See Supplementary Section 7 for more details.

## Data availability

The data that support the findings of this study are presented in the paper and Supplementary Information in graphical form. Datasets underlying the results presented in the paper are available at https://doi.org/10.5281/zenodo.17200209 (ref. 68), and from the authors upon reasonable request.

## Code availability

Numerical codes used in this study are based on MATLAB routines, as described in Methods and Supplementary Information. Parts relevant to reproduce the results of this work can be made available from the authors upon reasonable request.

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

## Acknowledgements

We acknowledge funding from Deutsche Forschungsgemeinschaft (Li 580/16-1, INST 184/163-1, INST 184/164-1 (C.L.), DE 3578/3-1 (A.D.S.), SFB 1372 'Magnetoreception and navigation in vertebrates' project number 395940726 (C.L. and A.D.S.) and SFB 1772 'Heterostrukturen aus Molekülen und zweidimensionalen Materialien' project A03 (A.K. and L.G.)). We thank the Niedersächsische Ministerium für Wissenschaft und Kultur (DyNano (C.L. and A.D.S.) and Wissenschaftsraum ElLiKo (C.L.)). M.S. acknowledges funding from BMFTR (NanoMatFutur FKZ: 13 N13637, tubLAN Q.0). A.C.F. acknowledges funding from the Graphene Flagship, ERC Grants Hetero2D, GIPT, EU Grants GRAP-X, CHARM, ELEGANCE, PIXEurope, EPSRC Grants EP/K01711X/1, EP/K017144/1, EP/N010345/1, EP/L016087/1, EP/V000055/1 and EP/X015742/1. G.C. acknowledges support from the EIC Pathfinder Open programme (QUONDENSATE, 101130384) and by the European Union's NextGenerationEU Programme with the IPHOQS Infrastructure (IR0000016, ID D2B8D520, CUP B53C22001750006) 'Integrated Infrastructure Initiative in Photonic and Quantum Sciences'.

## Author contributions

M.G., M.S., S.S., A.D.S., C.L., A.R.C., B.L.T.R., W.L. and G.S. designed and fabricated the sample. D.T., M.G. and T.Q. constructed the experimental set-ups and D.T., M.G. and D.C.L. performed the experiments. D.T. and M.G. analysed the experimental results. S.S. and M.S. performed the FDTD simulations. D.T. performed the density matrix simulations. L.G., M.R. and A.K. performed the analytical modelling of the exciton–plasmon coupling. C.L., G.C. and A.C.F. conceived the project. D.T., C.L. and M.G. wrote the paper. All authors contributed to discussions or gave feedback to the final paper.

## Funding

## Competing interests

G.C. discloses association with NIREOS, a company that commercializes the TWINS birefringent interferometer. All other authors declare no competing interests.

## Additional information

**Correspondence and requests for materials** should be addressed to Christoph Lienau.

