## [Peer Review File · Nature Nanotechnology]

Ultrafast transition from coherent to incoherent polariton nonlinearities in a hybrid 1L-WS2/plasmon structure

Corresponding Author: Professor Christoph Lienau

Version 0:

Reviewer comments:

Reviewer #1

(Comments for the Author)

This manuscript uses ultrafast two-dimensional electronic spectroscopy (2DES) to uncover coherent polariton dynamics in a hybrid monolayer (1L) WS₂/plasmonic nanostructure. Compared to an uncoupled WS₂ flake, they observe an over 20-fold, polarization-dependent enhancement of the optical nonlinearity and a rapid evolution of the 2DES spectra within ~70 fs. They relate these dynamics to a transition from coherent polaritons to incoherent excitations, unravel the microscopic optical nonlinearities, and show the potential of coherent polaritons for ultrafast all-optical switching.

The manuscript claims that the exciton-plasmon hybridization results in a 20-fold increase in optical nonlinearity as compared to the uncoupled 1L-WS₂ and record distinct changes in the 2DES lineshape within the first ~70 fs, marking the transition from coherent to incoherent polaritons and long-lived (>10 ps) dark states. The authors rationalize these observations by considering the hybridization of plasmons with bright and dark excitons and the effects of EID and Pauli blocking on the polariton resonances. Furthermore, by employing theoretical modelling, their studies allow distinguishing between coherent and incoherent polaritons in 2DES, shedding light on their nonlinear response and paving the way to new routes towards their applications.

The results presented are original and significant. The type of coupling between silver nanoslit arrays and 1 layer WS₂ with such high temporal evolution is very hard to record. Thus, it is an experimental masterpiece. There have been very few of these experiments to date. The 20 fold increase in coupling is unusual and of significance. 2DES is a very hard technique and they have achieved very high signal to noise ratio with clean lineshapes in a single layer material. It is also the appropriate and most powerful technique to study such processes. Finally, the authors have done the needed theoretical simulations required to provide the insights into the physical processes taking place. This is crucial, because such information is contained in the line shapes and how they develop with the population/wait time T . I read this part very carefully and their modeling and analysis is solid. There is one aspect of the peak structure that the theory does not reproduce, but this is not unusual in such experiments. They have included all the parameters that I would expect them to include. Expansion of the theoretical modeling to fully reproduce the 2DES data could be part of future studies.

The manuscript is very clear and well written. I have only minor suggestions for improvement, mostly typos and minor errors.

In the introduction the authors state:

“In such systems, strong coupling to plasmons may result in a hybridization of bright and momentum-dark excitons.”

In fact, one could further expand:

“Recent studies have shown that hybridization of dark and bright excitons does occur, and it can be tuned with external magnetic fields” and perhaps cite Mapara et al, Nano Lett. 2022, 22, 4, 1680–1687

Typos and grammatical errors:

Abstract: “to a few ten of femtoseconds” should be “to a few tens of femtoseconds”

In page 9 line 259: “rung” I think should be “rank” as second rank (hierarchy) of many-body Hamiltonian.

Page 11 line 332: When “decorating” an Ag nanoslit array, the word “decorating” does not make sense to me. I would say “when placing a 1L-TMD onto an Ag nanoslit array” or “when a 1D-TMD is placed onto an Ag nanoslit array”.

Reviewer #2

(Comments for the Author)

This manuscript reports an elegant and timely study of ultrafast polariton dynamics in a hybrid WS₂–plasmon nano-slit system using sub-10 fs two-dimensional electronic spectroscopy. The experimental approach is technically impressive: the authors implement high-resolution 2DES in reflection geometry, enabling direct access to coherent polariton oscillations that were previously inaccessible with conventional pump–probe techniques. The data reveal a more than 20-fold enhancement of nonlinear signals compared to bare WS₂ and provide clear evidence of a transition from coherent polariton Rabi oscillations to incoherent populations and dark states on a ~70 fs timescale. These results represent an important advance in the field, both methodologically and in terms of physical insight.

The work is further strengthened by its theoretical analysis, where a minimal coupled-oscillator density-matrix model incorporating bright and dark excitons is used to reproduce and interpret the experimental observations. This framework convincingly links the observed nonlinear line shapes to nonlinear microscopic mechanisms such as Pauli blocking and excitation-induced dephasing. The combined experimental and theoretical treatment provides a coherent and compelling picture of coherent–to–incoherent polariton dynamics in 2D semiconductors. Overall, this is a high-quality study that makes a strong contribution to the understanding of polariton nonlinearities and will be of broad interest to the community. I believe it should be published in Nature Nanotech, with some minor revisions that can be considered.

1. Can the author comment on how the coherence beating depends on detuning and light-matter coupling strength? For eg, recent theory [Chng et al, J. Phys. Chem. Lett. 15, 11773-11783 (2024)] based on Fermi's golden rule suggests that there will be certain scaling laws and even turnover behavior of coherence time. Or maybe due to the high cavity loss, none of these would be seen in the experiments?
2. The sentence of line 43, "The resulting spread of the plasmonic near field in momentum space enables coupling to momentum-dark excitons", Is the momentum dark exciton the same as an optically dark exciton (zero transition dipole moment)? If so, how can there be a coupling between the dark exciton and the plasmonic field?
3. A typo in line 87, "exfoliation" should be "exfoliated"
4. In Fig.3e, how is the dynamics of the signals recorded? Is it integrated over a small region around the points located, or is it the actual magnitude of the exact locations marked in Fig. 3a? Also, are these signals normalized with respect to the maximum amplitude of 2DES across the t_2 interval?
5. In line 197, does the "incoherent" refer to the steady state magnitude of each of the signals?
6. In line 221, I do not understand the coupling of what to SPP? This is talking about the coupling between X_A and SPP being stronger than X_B and SPP, right? Or is it talking about the interaction between the 2DES lasers and the system? It would be better for the reader if these were explained a little bit more clearly. Here, X_A and X_B should be explicitly mentioned if this is the case.
7. In line 248, the "P-resonance" should be defined first as "Plasmon resonance" as this terminology is not defined before.
8. In line 253, and also throughout the manuscript, it would be good if you consistently use the bra-ket notation throughout the paper for the different states.
9. In line 278, does this refer to Fig. 4e and Fig. 4g instead of the mentioned Fig. 4a?

Reviewer #3

(Comments for the Author)

This manuscript presents an experimental study of a TMD-plasmonic sample using a combination of transient absorption and multidimensional coherent spectroscopy. The results provide a careful and detailed measurement of the transition from the coherent to the incoherent regime for the nonlinear optical response. The interpretation of the measurements is supported by multi-level calculations that phenomenologically incorporate many-body phenomena, which are dominant. The paper is well written with a clear structure to help the reader follow a somewhat complex chain of analysis.

I believe this paper provides important new insight into the dynamics of hybrid TMD-metal plasmonic structures that show the presence of polariton states. As such I recommend publication in Nature Nanotechnology.

Prof. Dr. Christoph Lienau
Institut für Physik
Carl von Ossietzky Universität
26129 Oldenburg
Germany

Oldenburg, 26.09.2025

Dr. Alberto Moscatelli
Chief Editor
Nature Nanotechnology

Revisions for Manuscript NNANO-25072948 for Nature Nanotechnology

Dear Dr Moscatelli,

We are glad to hear the positive reviews for our manuscript, supporting publication in Nature Nanotechnology. In our revised manuscript we tried to address all concerns and made all relevant changes. Our respective changes are summarized below. We have also added the requested brief 250 character summary of our main findings.

We hope that the paper now meets all criteria for publication in Nature Nanotechnology. We also want to thank you for the editorial handling of our manuscript.

With my best regards,

Christoph Lienau

Brief summary of our findings

Strongly enhanced optical nonlinearities and an ultrafast transition from coherent to incoherent polariton excitations are demonstrated by coupling an atomically thin WS₂ layer to a plasmonic nanostructure using ultrafast multidimensional spectroscopy.

Response to the Reviewer's Reports

We want to thank the Reviewers for taking their time in reading and critically evaluating our manuscript. We are glad to read the positive reviews and that all three Reviewers support publication in Nature Nanotechnology.

In the following, we try to address all concerns of the Reviewers as good as possible. We mark the quotes of the Reviewer reports by blue text color and italic font.

Reviewer 1

This manuscript uses ultrafast two-dimensional electronic spectroscopy (2DES) to uncover coherent polariton dynamics in a hybrid monolayer (1L) WS₂/plasmonic nanostructure. Compared to an uncoupled WS₂ flake, they observe an over 20-fold, polarization-dependent enhancement of the optical nonlinearity and a rapid evolution of the 2DES spectra within ~70 fs. They relate these dynamics to a transition from coherent polaritons to incoherent excitations, unravel the microscopic optical nonlinearities, and show the potential of coherent polaritons for ultrafast all-optical switching.

The manuscript claims that the exciton-plasmon hybridization results in a 20-fold increase in optical nonlinearity as compared to the uncoupled 1L-WS₂ and record distinct changes in the 2DES lineshape within the first ~70 fs, marking the transition from coherent to incoherent polaritons and long-lived (>10 ps) dark states. The authors rationalize these observations by considering the hybridization of plasmons with bright and dark excitons and the effects of EID and Pauli blocking on the polariton resonances. Furthermore, by employing theoretical modelling, their studies allow distinguishing between coherent and incoherent polaritons in 2DES, shedding light on their nonlinear response and paving the way to new routes towards their applications.

The results presented are original and significant. The type of coupling between silver nanoslit arrays and 1 layer WS₂ with such high temporal evolution is very hard to record. Thus, it is an experimental masterpiece. There have been very few of these experiments to date. The 20 fold increase in coupling is unusual and of significance. 2DES is a very hard technique and they have achieved very high signal to noise ratio with clean lineshapes in a single layer material. It is also the appropriate and most powerful technique to study such processes. Finally, the authors have done the needed theoretical simulations required to provide the insights into the physical processes taking place. This is crucial, because such information is contained in the line shapes and how they develop with the population/wait time T . I read this part very carefully and their modeling and analysis is solid. There is one aspect of the peak structure that the theory does not reproduce, but this is not unusual in such experiments. They have included all the parameters that I would expect them to include. Expansion of the theoretical modeling to fully reproduce the 2DES data could be part of future studies.

The manuscript is very clear and well written. I have only minor suggestions for improvement, mostly typos and minor errors.

Reply: We thank the Reviewer for their time and for supporting publication in Nature Nanotechnology after minor revisions. We also want to thank the Reviewer for the nice words regarding our study.

In the introduction the authors state:

“In such systems, strong coupling to plasmons may result in a hybridization of bright and momentum-dark excitons.”

In fact, one could further expand:

“Recent studies have shown that hybridization of dark and bright excitons does occur, and it can be tuned with external magnetic fields” and perhaps cite Mapara et al, Nano Lett. 2022, 22, 4, 1680–1687

Reply: In the study by Mapara et al.¹ that is mentioned by the Reviewer, the authors use strong magnetic fields to hybridize spin-allowed bright and spin-forbidden dark excitons in monolayer WSe₂. Using a four-wave mixing technique, the coherent system evolution following impulsive interaction with an ultrashort laser pulse is recorded. This gives access to the dephasing time of the system and also to coherent quantum beats arising from couplings in the system. Using strong external magnetic fields, Mapara et al. hybridize the spin bright and dark excitons via spin tilting with in-plane magnetic fields. As a result, 100-fs quantum beats appear as a signature of the coherent magnetic field-induced coupling between the spin-bright and ~38 meV lower-lying spin-dark excitons. Additional effects on the dephasing time are observed which result from this hybridization.

In our WS₂-based sample, such spin-forbidden dark excitons are also expected to be present in a similar energy range, ~50 meV below the spin bright A exciton.² In plasmonic nanostructures, it has been shown that plasmonic near fields can indeed be used to couple to these spin dark excitons since they possess a small out-of-plane transition dipole moment.³

The dark excitons that we consider in our coupling to plasmonic near fields are momentum dark and lie outside of the light cone of far field radiation. In contrast to the spin dark excitons that are energetically detuned by ~50 meV,² a large density of states of momentum dark excitons is expected to be essentially in resonance with the momentum bright exciton or at slightly higher energies. We therefore expect the main effect of such spin dark excitons to be a part of the long-lived reservoir of dark states that are incoherently populated.

Following the suggestion of the Reviewer, we have now revised our introduction as followed:

Changes: On p. 2 we now added:

“For spin bright and dark excitons in monolayer WSe₂, such a hybridization can be induced by external magnetic fields.”

Typos and grammatical errors:

Abstract: “to a few ten of femtoseconds” should be “to a few tens of femtoseconds”

In page 9 line 259: “rung” I think should be “rank” as second rank (hierarchy) of many-body Hamiltonian.

Page 11 line 332: When “decorating” an Ag nanoslit array, the word “decorating” does not make sense to me. I would say “when placing a 1L-TMD onto an Ag nanoslit array” or “when a 1D-TMD is placed onto an Ag nanoslit array”.

Reply: We thank the Reviewer for pointing out these mistakes. We have corrected them accordingly.

Reviewer 2

This manuscript reports an elegant and timely study of ultrafast polariton dynamics in a hybrid WS₂-plasmon nano-slit system using sub-10 fs two-dimensional electronic spectroscopy. The experimental approach is technically impressive: the authors implement high-resolution 2DES in reflection geometry,

enabling direct access to coherent polariton oscillations that were previously inaccessible with conventional pump–probe techniques. The data reveal a more than 20-fold enhancement of nonlinear signals compared to bare WS₂ and provide clear evidence of a transition from coherent polariton Rabi oscillations to incoherent populations and dark states on a ~70 fs timescale. These results represent an important advance in the field, both methodologically and in terms of physical insight.

The work is further strengthened by its theoretical analysis, where a minimal coupled-oscillator density-matrix model incorporating bright and dark excitons is used to reproduce and interpret the experimental observations. This framework convincingly links the observed nonlinear line shapes to nonlinear microscopic mechanisms such as Pauli blocking and excitation-induced dephasing. The combined experimental and theoretical treatment provides a coherent and compelling picture of coherent–to–incoherent polariton dynamics in 2D semiconductors. Overall, this is a high-quality study that makes a strong contribution to the understanding of polariton nonlinearities and will be of broad interest to the community. I believe it should be published in Nature Nanotech, with some minor revisions that can be considered.

Reply: We thank the Reviewer for their time and for supporting publication in Nature Nanotechnology after minor revisions. We also would like to thank the Reviewer for the kind words.

1. Can the author comment on how the coherence beating depends on detuning and light-matter coupling strength? For eg, recent theory [Chng et al, J. Phys. Chem. Lett. 15, 11773-11783 (2024)] based on Fermi's golden rule suggests that there will be certain scaling laws and even turnover behavior of coherence time. Or maybe due to the high cavity loss, none of these would be seen in the experiments?

Reply: In our plasmonic nanostructure, detuning between the plasmon and WS₂ A exciton at ~2 eV can be achieved by varying the angle of incidence. For the angle of 3° used in the nonlinear experiments, we are in a regime of small positive detuning $\Delta = E_p - E_x \approx 15$ meV.

The theoretical study by Chng et al.⁴ that is highlighted by the Reviewer focuses on coherence properties of molecular cavity polaritons within a Holstein-Tavis-Cummings Model. Effects of the number of emitters N , their individual coupling strengths, and of detuning and cavity losses are investigated. Importantly, $N-1$ dark states are included in the model. The paper shows that for sufficiently strong collective coupling, polariton coherence can be increased (up to ~150 fs) compared to that of the uncoupled molecule (~15 fs). In the investigated regime, the limiting mechanism for polariton decoherence is a population transfer, mainly from the upper polariton, to the energetically lower-lying dark states. The interplay between the effects of this population transfer and cavity losses for the polariton decoherence is discussed. In case of a positive detuning Δ , it is shown that a turnover behavior occurs, leading to a maximum in the predicted polariton coherence time for a finite detuning value.

The reported key role of population transfer to dark states for the polariton coherence is very much in line with the findings of our study. We want to thank the Reviewer for bringing this to our attention. We think that the investigated systems are in fact quite different. Chng et al. study (non-interacting) molecules within a Tavis-Cummings model and take the role of molecular vibrations into account. In our case we discuss interacting semiconductor excitons in a coupled oscillator model and consider the coupling to vibrations only indirectly via the exciton dephasing in the Lindblad description.⁵ More importantly the considered cavity modes differ largely. The Q-factors used in the study by Chng et al. are 167 or higher. For our system, the Q factor is more on the order of $Q_{SPP} = E_{SPP}/(2\gamma_{SPP}) \approx 30$ (using $\gamma_{SPP} = 34$ meV and $E_{SPP} = 2.015$ eV).

In our system, the combination of low SPP Q factor and rapid population transfer into dark states limits the polariton coherence to less than 50 fs. This short decoherence makes it difficult to quantitatively investigate the appearance of the discussed turnover point. For such tests an extension of our work to

hybrid nanostructure with longer polariton decoherence times seems necessary. In principle, this may be achieved either by increasing the exciton-plasmon coupling strength or the Q factor. Plasmonic nano-slit arrays can in principle offer the required narrow linewidths and resulting larger Q factors on the order of 100-200.^{6,7}

Changes: We now added on p. 9:

“In our model, the $|D\rangle$ state population decays mainly due to dephasing-induced back transfer of population into the radiatively damped polaritons. **A recent theoretical study of molecular cavity polaritons also suggests that polariton decoherence arises from population transfer into collective dark states, mediated by molecular vibrations.**”⁴”

2. The sentence of line 43, “The resulting spread of the plasmonic near field in momentum space enables coupling to momentum-dark excitons”, Is the momentum dark exciton the same as an optically dark exciton (zero transition dipole moment)? If so, how can there be a coupling between the dark exciton and the plasmonic field?

Reply: The Reviewer addresses an important point: to be optically bright, two-dimensional excitons must obey the optical selection rules (non-vanishing dipole moment for orbital and spin transitions) and, at the same time, momentum conservation between the center-of-mass momentum q_{\parallel} (2D) and the 3D momentum k_0 of the incident light. The phrase *momentum-dark excitons* applies in our notation, if momentum conservation is not satisfied but orbital and spin transitions are allowed.

Therefore, momentum-dark excitons having $q_{\parallel} > k_0$ still exhibit a transition dipole moment (spin and orbital degree of freedom) identical to that of bright TMD excitons which obey $q_{\parallel} \approx 0$. Momentum dark excitons, in contrast, become only accessible once the momentum mismatch $q_{\parallel} > k_0$ is overcome, for instance by coupling to evanescent electric near-fields of nanostructures which generate a momentum distribution $q_{\parallel} \gg k_0$.

In more detail, the momentum conservation can be understood as follows:

Free propagating photons from the far field fulfill the dispersion relation,

$$\frac{\varepsilon\omega_0^2}{c^2} = k_0^2, \quad (1)$$

between energy $\hbar\omega_0$ and photon momentum $\hbar k_0$. Eq. (1) restricts the photon momentum to small values, i.e., the so-called *light cone*:

In a TMD monolayer, the in-plane translational invariance requires conservation of in-plane momentum between the absorbed photon and the excited exciton. This condition distinguishes TMD Wannier excitons from Frenkel excitons in, for example, J-aggregates.

Consequently, TMD Wannier excitons with center-of-mass momentum $q_{\parallel} > k_0 = \sqrt{\varepsilon} \frac{\omega_0}{c} \approx 0.018/\text{nm}$ (for $\varepsilon = (n_{\text{Al}_2\text{O}_3})^2 = 3.1$ and $\hbar\omega = 2$ eV) cannot be excited by far-field photons, since such transitions are forbidden by momentum mismatch (in contrast to spin-forbidden excitons with zero transition dipole moment, as mentioned by the Reviewer).

Changes: We now revised the text on p. 1:

“The resulting spread of the plasmonic near field in momentum space enables coupling to momentum-dark excitons, which – **due to momentum mismatch** – usually cannot be excited by far-field light.”

3. A typo in line 87, “exfoliation” should be “exfoliated”

Reply: We corrected the typo as pointed out by the Reviewer.

4. In Fig.3e, how is the dynamics of the signals recorded? Is it integrated over a small region around the points located, or is it the actual magnitude of the exact locations marked in Fig. 3a? Also, are these signals normalized with respect to the maximum amplitude of 2DES across the t_2 interval?

Reply: The signals plotted in Fig. 3e are dynamics along the waiting time T taken at the positions marked in Fig. 3a as the average within a ± 5 meV square. This averaging region is sufficiently small to not introduce any smearing out of spectral features. The amplitude in A_{2D} of the 2DES signal is an arbitrary unit that results after the Fourier transform of the coherence time scan. No normalization is applied to the pump-probe and 2DES data at any point of data analysis. The amplitude displayed in Fig. 3e is directly corresponding to the amplitude at the marked positions in the 2DES maps as marked by the color code.

Changes: To improve the clarity we adapted the caption of Fig. 3 on p. 7:

“e: Waiting time dynamics of the 2DES peaks 1-5 taken at the positions marked in (a) within a ± 5 meV window.”

5. In line 197, does the “incoherent” refer to the steady state magnitude of each of the signals?

Reply: We are not totally sure whether we interpret the question correctly. In Fig. 3e, we see, for waiting times of less than 50 fs, partly oscillatory changes in the amplitudes of the 2DES peaks. This dynamics reflect the excitation of a coherent superposition of different polariton eigenstates, as discussed in the paper. For waiting times beyond 50 fs, the peak amplitudes show a monotonous and slow decay. Quantitatively, we can describe it by a 3-exponential decay with time constants of 0.5 ps, 3.1 ps and 28 ps, as shown in Fig. R1. As such, the amplitude of the signals in Fig. 3e for waiting times up to 90 fs can be considered as “quasi-static”. At this moment in time the polariton decoherence and population transfer into dark states has destroyed the coherent superposition of different polariton eigenstates. The density matrix of the system then effectively is a statistical ensemble of incoherent polariton eigenstates and long-lived dark exciton states without significant coupling to the plasmonic fields. We therefore wrote that “the temporal evolution of the 2DES marks the transition from impulsively-excited, short-lived coherent polaritons to incoherent polariton excitations and long-lived dark states, demonstrated here for a 1L-TMD/metal hybrid.”

Changes: We now state that the incoherent dark state population lives for >20 ps and refer to a new Figure S19 in the Supporting Information that quantifies the decay on longer time scales:

“...transition from impulsively-excited, short-lived coherent polaritons to incoherent polariton excitations and long-lived (>20 ps, Fig. S19) dark states, demonstrated here for a 1L-TMD/metal hybrid.”

Figure R1: New Figure S19, showing slow dynamics of the sample with a 3 exponential fit with time constants of 0.5 ps, 3.1 ps and 28 ps.

6. In line 221, I do not understand the coupling of what to SPP? This is talking about the coupling between X_A and SPP being stronger than X_B and SPP, right? Or is it talking about the interaction between the 2DES lasers and the system? It would be better for the reader if these were explained a little bit more clearly. Here, X_A and X_B should be explicitly mentioned if this is the case.

Reply: In line 221 on p. 8 we state “Importantly, far-field coupling to the SPP resonance is much stronger than to $|X_B\rangle$.” This statement refers indeed to the coupling between the 2DES lasers (“far-field”) and the system. To make this more clear, we revised the statement accordingly.

Changes: on p. 8 we now write:

“Importantly, optical absorption of the SPP resonance is much stronger than that of $|X_B\rangle$.”

7. In line 248, the “P-resonance” should be defined first as “Plasmon resonance” as this terminology is not defined before.

Reply: Indeed, we missed to define P as the plasmon. Taking also into account to use a bra-ket notation as mentioned by the Reviewer in the next point, we also now directly refer to the plasmon state as $|P\rangle$.

Changes: on p. 9 we now write:

“...mainly couples to the plasmon resonance $|P\rangle$ and...”

8. In line 253, and also throughout the manuscript, it would be good if you consistently use the bra-ket notation throughout the paper for the different states.

Reply: We thank the Reviewer for pointing out that we missed to use proper bra-ket notation at several places. Bra-ket notation is now used throughout the manuscript when referring to states.

9. In line 278, does this refer to Fig. 4e and Fig. 4g instead of the mentioned Fig. 4a?

Reply: In line 278 on p. 9 we write “Due to EID, the linewidth of the optically allowed transitions between $|UP\rangle$ and $|LP\rangle$ and the $2Q$ states (Fig. 4a) is slightly larger than that of the GSB/SE band.” Here, we refer to Fig. 4a regarding the optically allowed transitions that are marked in the level scheme and not to the 2DES maps shown in Fig. 4e,g.

Changes: To make clearer that we want to refer to the level scheme in Fig. 4a we revised the sentence accordingly on p. 9:

“...2Q states (**level scheme in Fig. 4a**) is slightly larger...”

Reviewer 3

This manuscript present an experimental study of a TMD-plasmonic sample using a combination of transient absorption and multidimensional coherent spectroscopy. The result provide a careful and detailed measurement of the transition from the coherent to the incoherent regime for the nonlinear optical response. The interpretation of the measurements are supported by multi-level calculations that phenomenologically incorporate many-body phenomena, which are dominant. The paper is well written with a clear structure to help the reader follow a somewhat complex chain of analysis.

I believe this paper provide important new insight into the dynamics of hybrid TMD-metal plasmonic structures that show the presence of polariton states. As such I recommend publication in Nature Nanotechnology.

Reply: We thank the Reviewer for their time and for supporting publication in Nature Nanotechnology and for the kind words regarding our study.

References

- (1) Mapara, V.; Barua, A.; Turkowski, V.; Trinh, M. T.; Stevens, C.; Liu, H.; Nugera, F. A.; Kapuruge, N.; Gutierrez, H. R.; Liu, F.; Zhu, X.; Semenov, D.; McGill, S. A.; Pradhan, N.; Hilton, D. J.; Karaiskaj, D. Bright and Dark Exciton Coherent Coupling and Hybridization Enabled by External Magnetic Fields. *Nano Letters* **2022**, *22* (4), 1680-1687.
- (2) Molas, M. R.; Faugeras, C.; Slobodeniuk, A. O.; Nogajewski, K.; Bartos, M.; Basko, D. M.; Potemski, M. Brightening of dark excitons in monolayers of semiconducting transition metal dichalcogenides. *2d Materials* **2017**, *4* (2), 021003.
- (3) Zhou, Y.; Scuri, G.; Wild, D. S.; High, A. A.; Dibos, A.; Jauregui, L. A.; Shu, C.; De Greve, K.; Pistunova, K.; Joe, A. Y.; Taniguchi, T.; Watanabe, K.; Kim, P.; Lukin, M. D.; Park, H. Probing dark excitons in atomically thin semiconductors via near-field coupling to surface plasmon polaritons. *Nature Nanotechnology* **2017**, *12* (9), 856-860.
- (4) Chng, B. X. K.; Ying, W.; Lai, Y.; Vamivakas, A. N.; Cundiff, S. T.; Krauss, T. D.; Huo, P. Mechanism of Molecular Polariton Decoherence in the Collective Light–Matter Couplings Regime. *The Journal of Physical Chemistry Letters* **2024**, *15* (47), 11773-11783.
- (5) Gallego-Valencia, D.; Mewes, L.; Feist, J.; Sanz-Vicario, J. L. Coherent multidimensional spectroscopy in polariton systems. *Physical Review A* **2024**, *109* (6), 063704.
- (6) Timmer, D.; Gittinger, M.; Quenzel, T.; Stephan, S.; Zhang, Y.; Schumacher, M. F.; Lützen, A.; Silies, M.; Tretiak, S.; Zhong, J.-H.; De Sio, A.; Lienau, C. Plasmon mediated coherent population oscillations in molecular aggregates. *Nature Communications* **2023**, *14* (1), 8035.
- (7) Ropers, C.; Park, D.; Stibenz, G.; Steinmeyer, G.; Kim, J.; Kim, D.; Lienau, C. Femtosecond light transmission and subradiant damping in plasmonic crystals. *Physical Review Letters* **2005**, *94* (11), 113901.